# RiEMann: Near Real-Time $SE(3)$-Equivariant Robot Manipulation without Point Cloud Segmentation

**Chongkai Gao**
National University of Singapore
gaochongkai@u.nus.edu

**Zhengrong Xue**
Tsinghua IIIS
xzr23@mails.tsinghua.edu.cn

**Shuying Deng**
Tsinghua University
starrism1412@gmail.com

**Tianhai Liang**
Tsinghua University
tianhailiang.cn@gmail.com

**Siqi Yang**
Tsinghua University
yang-sq21@mails.tsinghua.edu.cn

**Lin Shao**
National University of Singapore
linshao@nus.edu.sg

**Huazhe Xu**
Tsinghua IIIS
Shanghai AI Lab
Shanghai Qi Zhi Institute
huazhe_xu@mail.tsinghua.edu.cn

**Abstract:** We present RiEMann, an end-to-end near Real-time SE(3)-Equivariant Robot Manipulation imitation learning framework from scene point cloud input. Compared to previous methods that rely on descriptor field matching, RiEMann directly predicts the target actions for manipulation without any object segmentation. RiEMann can efficiently train the visuomotor policy from scratch with 5 to 10 demonstrations for a manipulation task, generalizes to unseen SE(3) transformations and instances of target objects, resists visual interference of distracting objects, and follows the near real-time pose change of the target object. The scalable SE(3)-equivariant action space of RiEMann supports both pick-and-place tasks and articulated object manipulation tasks. In simulation and real-world 6-DOF robot manipulation experiments, we test RiEMann on 5 categories of manipulation tasks with a total of 25 variants and show that RiEMann outperforms baselines in both task success rates and SE(3) geodesic distance errors (reduced by 68.6%), and achieves 5.4 frames per second (fps) network inference speed.

**Keywords:** SE(3)-Equivariance, Manipulation, Imitation Learning

## 1 Introduction

Learning from demonstrations is an effective and convenient mechanism for visual robot manipulation tasks [1, 2]. However, most current algorithms for learning from demonstrations suffer from low data efficiency and generalization ability to new situations. For example, current visual imitation learning algorithms require around 100 demonstrations [3, 4, 5] to learn a simple manipulation task such as picking up a mug and placing it on a rack, and cannot generalize to new object poses beyond the training distribution. Although previous works have tried to tackle these problems by data augmentation [6, 7] or contrastive learning on demonstrations [8, 9, 10], they rely on task-specific domain knowledge and have no algorithmic guarantees to generalize to unseen object poses.

Exploiting symmetries in the 3D world can improve the sample efficiency and generalization ability of robot learning algorithms [11]. Roto-translation equivariance is one of the most common types

8th Conference on Robot Learning (CoRL 2024), Munich, Germany.

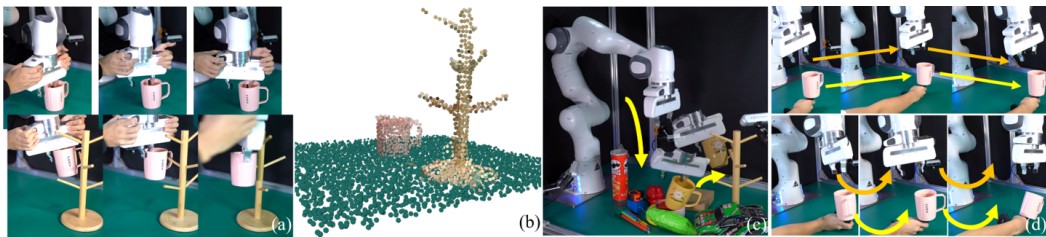

Figure 1: Overview of RiEMann. (a) Given 5 to 10 demonstrations of restricted object poses (The mug remains standing and only rotates in 90 degrees around the z-axis) of the task *Mug on Rack* and (b) with the full scene point could as input without segmentation, (c) RiEMann can generalize to local SE(3)-equivariant transformations of target objects, to new instances of target objects, be robust to distracting objects, and (d) has the near real-time following ability of target objects.

of symmetry for robot manipulation tasks. It denotes that robot actions can transform with the same SE(3) transformations as the target objects. Pioneering works like NDFs [12, 13] and EDFs [14, 15] propose to first learn an SE(3)-equivariant object descriptor field with SE(3)-equivariant backbones [16, 17, 18] and then output robot actions with energy-based optimization to match different descriptor fields. They achieve SE(3)-equivariance on pick-and-place tasks such as picking up the mug to the rack [14, 12, 13, 15]. However, these methods share the same limitations of the field-matching mechanism: 1) the matching process is time-consuming, preventing it from being a real-time method; 2) the matching process can only yield one SE(3) transformation, which only works on pick-and-place tasks and precludes its application to articulated object manipulation tasks.

Recently, some works have been trying to solve these issues. Fourier Transporter [19] and TAX-Pose [20] try to tackle the first issue by cross correlation of Fourier transformation on SO(3) [19] or differential SVD [20] to replace the time-consuming optimization process, but they either require exorbitant computation to discretize the SO(3) group at a high resolution, or require a pre-trained encoder on an extra well-segmented dataset. EquivAct [21] tries to tackle the second problem by directly predicting robot actions with neural networks, but they require well-segmented point clouds and can only predict 3-DOF actions rather than 6-DOF actions. These works inspire us to design an end-to-end SE(3)-equivariant policy that enables 6-DOF manipulation tasks without segmentation to replace the field matching process, as well as an efficient network structure and training algorithm to alleviate complexity brought by equivariant backbones to improve the inference speed.

In this work, we present RiEMann, the first near Real-time SE(3)-Equivariant robot Manipulation framework from point cloud inputs without any segmentation. RiEMann leverages local SE(3)-equivariant backbones [17] and learns a policy that directly outputs SE(3)-equivariant actions. For the 6-DOF action space parameterization problem, RiEMann uses an SE(3)-invariant vector field as the target point affordance map for translational actions, and uses three SE(3)-equivariant vector fields as three bases of the target rotation matrix as well as a Gram-Schmidt orthogonalization operation for the rotational actions. This action space can also be extended to articulated object manipulation tasks by adding an SE(3)-equivariant directional action. Theoretically, we prove that other common parameterizations for the SO(3) group, such as axis-angle, quaternion, and Euler angle, are not trainable SE(3)-equivariant representations. For the computational complexity problem, RiEMann firstly learns an SE(3)-invariant saliency map on the input scene point cloud to extract the region of interest, then trains the main SE(3)-equivariant policy on the extracted point cloud. Since the equivariant policy consumes the majority of computing and memory resources, reducing the number of points to be processed can significantly decrease computational and memory burden.

In 5 manipulation tasks with a total of 25 different task settings from simulation and real-world experiments, we demonstrate that RiEMann can solve various manipulation tasks with only 5 to 10 demonstrations for each task, generalize to unseen poses and instances of target objects, resist visual interference of distracting objects, have the near real-time (**5.4** FPS) inference speed as illustrated in Figure 1, and outperforms baselines on both success rates as well as the SE(3) geodesic distance errors (reduced the geodesic error by **68.6%**). In summary, our main contributions are as follows:

- We propose the first end-to-end SE(3)-equivariant visuomotor policy learning framework for 6-DOF manipulation tasks by our specially designed SE(3)-equivariant action space;
- We design a saliency map network together with position and orientation networks as the policy model to achieve efficient SE(3)-equivariant training and near real-time inference;
- We achieve state-of-the-art success rates and pose estimation accuracy on a set of pick-and-place tasks under various settings, and we are the first to perform SE(3)-equivariant articulated object manipulation tasks through end-to-end training.

## 2 Related Works

### 2.1 Group Equivariant Neural Networks

Exploiting groups of symmetry pervading in data and incorporating them into deep neural networks has been the focus of many studies since it can improve generalization and data efficiency. Equivariant neural networks, which are first introduced into CNNs [22], extract symmetries from various kinds of data. According to Bekkers [11], Han et al. [23], most equivariant models can be divided into two categories: regular group representation networks and steerable group representation networks. The former seeks to define equivariant convolution filters [22, 24], attention mechanism [25], or message passing mechanism [26] as functions on groups, while the latter uses irreducible group representations [27] with spherical harmonics as an equivariant basis to perform message passing [28, 29, 17, 18]. Some other works [16] design special non-linear kernels to achieve equivariance on the SO(3) group. The theory of equivariant networks on homogeneous spaces is formalized in Cohen et al. [30] with the vector bundle theory.

### 2.2 Equivariant Robot Manipulation

Researchers have been exploring equivariant models for robotic manipulation tasks to improve generalizability and sample efficiency. Pioneering works [31, 32, 33, 34, 35, 36] learn equivariant representations of objects or the scene with equivariant networks to get SO(2)- or SE(2)-equivariance for desktop manipulation tasks with imitation learning or reinforcement learning. NDFs [12, 13, 21] and TAX-Pose [20] leverages Vector Neurons [16] or SO(3)-data augmentation to acquire SE(3)-equivariant category-level object representations from point cloud of objects for downstream imitation learning, while EDFs [14, 15] uses SE(3)-Transformer [17] and Equiformers [18] to directly learn SE(3)-equivariant representations from point clouds of the scene. Huang et al. [19] leverages discrete approximation of steerable representation of SO(3) and uses cross correlation to tackle the optimization problem, but still requires huge computation for a high-resolution discretization. In this work, we leverage SE(3)-transformers [17] and design an end-to-end learning paradigm to predict target actions from scene point cloud input in near real-time.

## 3 Background and Problem Formulation

### 3.1 Problem Formulation

Let a colored point cloud with $N$ points be $\mathbf{P} = \{(x_1, c_1), \cdots, (x_N, c_N)\} \in \mathbb{R}^{N \times 6}$, where $x_i \in \mathbb{R}^3$ is the position and $c_i \in \mathbb{R}^3$ is the RGB color of the $i$-th point. For a manipulation task $\mathcal{T}$, a policy $f_\theta$ parameterized by $\theta$ is trained to predict the target pose of robot end-effector $\mathbf{T} = \{\mathbf{R}, \mathbf{t}\} \in SE(3)$ with a set of demonstrations $\mathcal{D} = \{(\mathbf{P}_i, \mathbf{T}_i)\}_{i=1}^m$ that consists of $m$ pairs, where $\mathbf{R} \in SO(3)$ is the 3D rotation action and $\mathbf{t} \in \mathbb{R}^3$ is the 3D translation action. We use $\widehat{\mathbf{T}} = f_\theta(\mathbf{P})$ to denote the predicted output. We assume a motion planner with collision avoidance is used to execute the action $\mathbf{T}$, as in Shridhar et al. [37], Simeonov et al. [12], Ryu et al. [14]. The trained policy will be tested on unseen target object poses, distracting objects, and new object instances, which relies on the following local SE(3)-equivariance property of our policy.

Let $g \in SE(3)$ be an element of the Special Euclidean Group in three dimensions (SE(3)) and $T_g : \mathcal{X} \to \mathcal{X}$ be a transformation of $g$ on a space $\mathcal{X}$. A function $f : \mathcal{X} \to \mathcal{Y}$ is called SE(3)-equivariant if there exists a transformation $S_g : \mathcal{Y} \to \mathcal{Y}$ such that:

$$S_g[f(x)] = f(T_g x), \quad \forall g \in SE(3), x \in \mathcal{X}. \tag{1}$$

In our robot manipulation tasks, $f$ is the trained policy, $\mathcal{X}$ is the space of the scene point cloud $\mathbf{P}$, $\mathcal{Y}$ is the space of target pose $\mathbf{T}$, $T_g$ is the space of any SE(3) transformation $T_g = \{\mathbf{R}, \mathbf{t}\}$ on $\mathbf{P}$, and $S_g$ is the SE(3) transformation on the target pose $\mathbf{T}$ and we set $S_g = T_g$ following [30, 11, 38].

Local SE(3)-equivariance [14, 15] refers that $T_g$ can be applied on only part of the input scene point cloud $\mathbf{P}$ rather than the whole scene input, as illustrated in Figure 1. This is more practical for robot manipulation tasks, in which we want the policy to be only equivariant to the target objects. Local equivariance can be achieved by equivariant networks with local mechanisms. In this work, we choose to use SE(3)-transformers [17] that belong to the latter category, as introduced below.

### 3.2 Group Representations and SE(3)-Transformers

$T_g$ and $S_g$ above are called *group representations*. Formally, a group representation $\mathbf{D}$ of a group $G$ is a map from group $G$ to the set of $N \times N$ invertible matrices $GL(N)$, and it satisfies $\mathbf{D}(g)\mathbf{D}(h) = \mathbf{D}(gh), \forall g, h \in G$. Specifically, any representation of SO(3) group $\mathbf{D}(\mathbf{R})$ for $\forall \mathbf{R} \in SO(3)$ can be block-diagonalized as the direct sum of orthogonal Wigner-D matrices $\mathbf{D}_l(\mathbf{R}) \in \mathbb{R}^{(2l+1) \times (2l+1)}$. Vectors transforming according to $\mathbf{D}_l(\mathbf{R})$ are called type-$l$ vectors [11]. Type-0 vectors are invariant under rotations ($\mathbf{D}(\mathbf{R}) = \mathbf{I}$) and type-1 vectors (3d space vectors) rotate according to 3D rotation matrices. Note, type-$l$ vectors have length $2l + 1$. Given a point cloud $\mathbf{P}$, we can assign a type-$l$ vector to each point $x \in \mathbf{P}$ to get a type-$l$ vector field $\mathbf{f}_l : \mathbb{R}^3 \to \mathbb{R}^{2l+1}$. Same or different type-$l$ vector fields can be stacked to get a complex vector field.

SE(3)-transformers [17] are neural networks that map point cloud vector fields to point cloud vector fields with the same point number, and they are designed to achieve *local SE(3)-equivariance* for any learned type-$l$ field:

$$\mathbf{D}_l(\mathbf{R})\mathbf{f}_l(x) = \mathbf{f}_l(Tx), \forall x \in \mathbf{P}, \forall T = \{\mathbf{R}, \mathbf{t}\} \in SE(3). \tag{2}$$

To leverage this property for robot manipulation tasks, we need to design special input and output vector fields to satisfy task requirements, as discussed in the following section.

## 4  SE(3)-Equivariant Robot Manipulation

Previous works like NDFs [12, 13] and EDFs [14, 15] output several type-0 or higher type-$l$ vector fields as the *object descriptor* for field-matching to get the desired robot action. In contrast, we aim to re-design the outputted vector fields and directly use them as *actions*. This can transform the generative modeling problem of the descriptor field into a supervised learning problem that enables higher training precision and a faster inference speed.

### 4.1 Design Choices of Vector Fields

There are two main requirements for the design of input and output vector fields: a) all the input (scene point cloud) and output (robot actions) can be incorporated into the vector fields; b) the predicted actions parameterized by output vector fields are SE(3)-equivariant to the input transformation. For the input design, since SE(3)-transformer is actually T(3)-invariant, we should only use the T(3)-invariant features, or visual features, as the input, to ensure T(3)-invariance. Thus here we only use color information for input. RGB information can be seen as 3 type-0 vectors, so our input is a direct sum of three vector fields:

$$\mathbf{f}_{in}(x) = \bigoplus_{i=1}^{3} \mathbf{f}_0^i(x), \forall x \in \mathbf{P}. \tag{3}$$

For the output, our model predicts the end-effector target pose $\mathbf{T} = \{\mathbf{R}, \mathbf{t}\} \in SE(3)$, which can be decomposed into target position $\mathbf{t}$ and target rotation $\mathbf{R}$. For target position $\mathbf{t}$, we cannot directly output the 3D target position vector since SE(3)-transformers are only able to predict T(3)-invariant features. Thus we choose to learn one type-0 vector field as an affordance map [39, 40] on point clouds, and use softmax on the affordance map to perform weighted sum on all point positions to get the translational action $\hat{\mathbf{t}}$. We also assume an offset $\mathbf{t}_o$ is provided if the target position is out of the convex hull of the object point cloud, and in this case the output is $\hat{\mathbf{t}} + \mathbf{t}_o$.

For the target orientation $\mathbf{R}$, people usually use Euler angle, quaternion, rotation matrix, or axis-angle as robot action parameterization. Although all these pose parameterizations can satisfy the first requirement (representing output action information), we need to ensure they are SE(3)-equivariant to the input transformation after parameterizing them with different type-$l$ vectors. We answer this question with the following theorems. All proofs are in Appendix A.3.

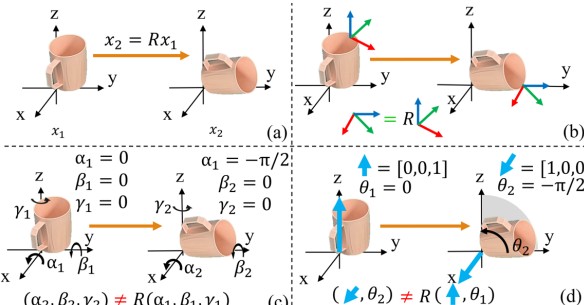

(a)   (b)

$(\alpha_2, \beta_2, \gamma_2) \neq R(\alpha_1, \beta_1, \gamma_1)$   (c)   $(\nearrow, \theta_2) \neq R(\uparrow, \theta_1)$   (d)

Figure 2: Illustration of different 3D rotation representations under type-$l$ parameterizations. (a) The initial point cloud $x_1$ is transformed to $x_2$ with 3D rotation $R$; (b) Using three type-1 vectors to represent a Rotation Matrix. It transforms with the same transformation $R$ as the input; (c) (d) Using three type-0 vectors to represent Euler Angles, and using one type-1 vector and one type-0 vector to represent Axis-angle. They cannot be transformed with the same transformation $R$ as the input, so they are not SE(3)-equivariant parameterizations.

**Theorem 1** *Rotation matrices, represented by three type-1 vectors, are SE(3)-equivariant vector field parameterization.*

**Theorem 2** *There is no SE(3)-equivariant vector field parameterization for Euler angle, quaternion, and axis-angle.*

The intuitions behind these theorems are illustrated in Figure 2. In short, rotation matrices, represented by three type-1 vectors, are the only SE(3)-equivariant rotation representation to the input transformation.

In summary, RiEMann learns one type-0 vector field (target point heatmap) for target position $\mathbf{t}$ and three type-1 vector fields for target rotation $\mathbf{R}$, which is a direct sum of 4 vector fields:

$$\mathbf{f}_{out}(x) = \mathbf{f}_0(x) + \bigoplus_{j=1}^{3} \mathbf{f}_1^j(x), \forall x \in \mathbf{P}. \tag{4}$$

As comparisons, EquivAct [21] uses one type-1 vector field as output (end-effector velocity in Cartesian space), which only supports 3-DOF manipulation tasks. Instead, RiEMann supports 6-DOF manipulation tasks. The next question is how to use networks to implement equation 3 and 4 efficiently and transform the output vector fields to a single output action while ensuring SE(3)-equivariance.

### 4.2 Network Design

The first problem of leveraging equivariant networks for scene point cloud input is the heavy computational and memory cost. A scene point cloud for robot manipulation tasks usually contains thousands to tens of thousands of points (in our case, 8192 points), and end-to-end learning on such a large number of points brings huge burdens for GPU memory and a long training time. For example, EDFs [14] only support batch size = 1 for training on an NVIDIA RTX3090 GPU. Meanwhile, the training difficulty also increases since most of the points are not informative. In this work, we employ an SE(3)-transformers $\phi(x)$ to output one type-0 vector field $\mathbf{f}_s(x) = \mathbf{f}_0(x), x \in \mathbf{P}$ for each point as a *saliency map*, and extract a relatively small region of point cloud $\mathbf{B}_{ROI}$ with a predefined radius $r_1$ centered on the point with the highest value, as illustrated in the left part of Figure 3.

With $\mathbf{B}_{ROI}$ as input, we then employ another two SE(3)-transformer networks to learn a translational action network $\psi_1(x)$ and an orientation network $\psi_2(x)$ for $\forall x \in \mathbf{B}_{ROI}$ as policy networks for action prediction. For the translational part, $\psi_1$ outputs one type-0 affordance $\mathbf{f}_t(x) = \mathbf{f}_0(x), x \in \mathbf{B}_{ROI}$. We then perform SoftMax on $\mathbf{f}_t(x)$ as weight and multiply them to the positions of all points of $\mathbf{B}_{ROI}$ to get the output translational action $\hat{\mathbf{t}}$. For the orientation part, $\psi_2$ outputs three type-1 vector fields $\mathbf{f}_R = \bigoplus_{i=1}^{3} \mathbf{f}_1^i(x), x \in \mathbf{B}_{ROI}$. Then we select the points from a smaller region that are centered on $\hat{\mathbf{t}}$ with a radius $r_2$, and perform mean pooling on these points to get the output orientation action $\hat{\mathbf{R}}$, as in the right side of Figure 3. During training, we use the translation action $\mathbf{t}$ in

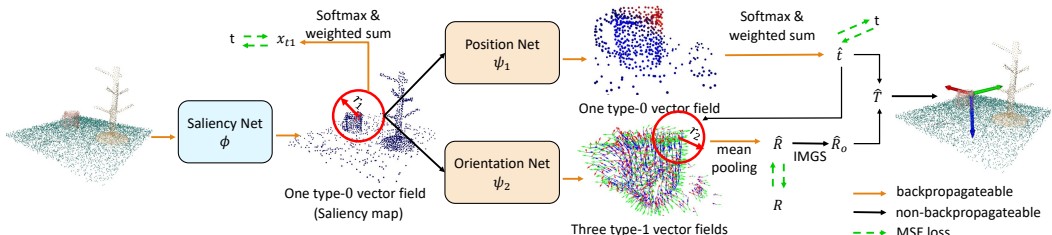

Figure 3: Pipeline of RiEMann. A type-0 saliency map is firstly outputted by an SE(3)-invariant backbone $\phi$ to get a small point cloud region $\mathbf{B}_{ROI}$, and an SE(3)-equivariant policy network that contains $\psi_1$ and $\psi_2$ predicts the action vector fields on the points of $\mathbf{B}_{ROI}$. Finally, we perform softmax, region mean pooling, and IMGS to get the target action $\mathbf{T}$.

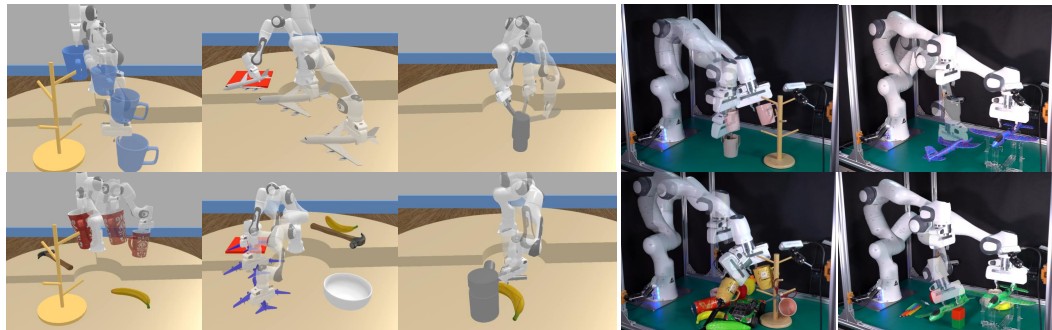

Figure 4: Simulation and real-world environments. Top: the training environment settings of simulation tasks and real world tasks. Bottom: the *ALL* testing case of tasks, where the target object is a new instance and in a new pose, and distracting objects are added in the environments.

the demonstrations as supervision for both $\phi(x)$ and $\psi_1(x)$, and three orientation axes of $\mathbf{R}$ to train $\psi_2(x)$. The training algorithm of RiEMann is shown in Algorithm 1 in Appendix A.1. Note the three type-1 vectors from $\psi_2$ are not necessarily orthogonal to each other, in which case they cannot form a legal rotation matrix. Here we perform Iterative Modified Gram-Schmidt orthogonalization (IMGS) [41] to get an orthogonal matrix $\widehat{\mathbf{R}}_o$ for actual robot control, as in Appendix A.2.

For articulated object manipulation, we assume that given a target pose $\mathbf{T} = \{\mathbf{R}, \mathbf{t}\} \in SE(3)$ and a target direction $\mathbf{d} \in \mathbb{R}^3$, the robot can accomplish the task by first going to the target pose $\mathbf{T}$ just as done in pick-and-place tasks, and then moving along the target direction $\mathbf{d}$ while keeping the orientation not changed, as illustrated in Figure 6. We expend the action space with another type-1 vector field on the orientation network $\psi_2$: $\mathbf{f}_R = \bigoplus_{i=1}^{4} \mathbf{f}_1^i(x), x \in \mathbf{B}_{ROI}$. The final output directional action $\hat{\mathbf{d}}$ is also calculated through mean pooling on points in the radius $r_2$. Note the policy needs to continuously predict the output direction during the opening process, which shows that RiEMann can capture the local SE(3)-equivariance of the handle part of the faucet and has near real-time inference speed. Detailed implementation details of our model are listed in Appendix A.4

## 5 Experiments

We systematically evaluate RiEMann in both simulation and real-world experiments. This evaluation includes 3 types of manipulation tasks in simulation environments and 2 types of manipulation tasks in the real world, with 5 different settings and 10 demonstrations for each task. The ablation studies are in Appendix A.9.

### 5.1 Simulation Environments and Tasks

We build all simulation manipulation environments based on Sapien [42] for point-cloud-based manipulation. We build a ring-table experiment platform with uneven tabletop for all tasks in simulation, as shown in Figure 4. We put 6 RGBD cameras around the table to fuse their captures to get the point cloud input, and downsample the scene point cloud to 8192 points. For each manipulation task, we collect demonstrations and train our policy under the *training setting* (**T**), and test the

Table 1: Success rates of different tasks in simulation. Evaluated under 20 random seeds.

| Method | Mug on Rack | | | | | Plane on Shelf | | | | | Turn Faucet | | | | |
| --- | --- | --- | --- | --- | --- | --- | --- | --- | --- | --- | --- | --- | --- | --- | --- |
| | T | NI | NP | DO | ALL | T | NI | NP | DO | ALL | T | NI | NP | DO | ALL |
| PerAct [37] | 0.85 | 0.00 | 0.70 | 0.00 | 0.00 | 0.90 | 0.00 | 0.80 | 0.00 | 0.00 | 0.45 | 0.00 | 0.50 | 0.00 | 0.00 |
| R-NDF [13] | 0.00 | 0.00 | 0.00 | 0.00 | 0.00 | 0.00 | 0.00 | 0.00 | 0.00 | 0.00 | n/a | n/a | n/a | n/a | n/a |
| EDF [14] | 1.00 | 0.85 | 1.00 | 0.95 | 0.80 | 0.90 | 0.75 | 0.80 | 0.85 | 0.70 | n/a | n/a | n/a | n/a | n/a |
| D-EDF [15] | 1.00 | 0.85 | 0.95 | 0.95 | 0.75 | 1.00 | 0.80 | 0.95 | 0.95 | 0.75 | n/a | n/a | n/a | n/a | n/a |
| RiEMann (Ours) | 1.00 | 0.90 | 0.95 | 1.00 | 0.85 | 1.00 | 0.90 | 1.00 | 1.00 | 0.90 | 1.00 | 0.75 | 1.00 | 1.00 | 0.65 |

Table 2: SE(3) Geodesic distances of tasks in simulation. Evaluated under 20 random seeds.

| Method | Mug on Rack | | | | | Plane on Shelf | | | | | Turn Faucet | | | | |
| --- | --- | --- | --- | --- | --- | --- | --- | --- | --- | --- | --- | --- | --- | --- | --- |
| | T | NI | NP | DO | ALL | T | NI | NP | DO | ALL | T | NI | NP | DO | ALL |
| PerAct [37] | 0.393 | 4.086 | 0.698 | 4.166 | 4.375 | 0.431 | 4.806 | 0.469 | 4.752 | 4.993 | 0.457 | 4.365 | 0.382 | 4.218 | 4.039 |
| R-NDF [13] | 4.855 | 4.298 | 4.178 | 4.509 | 4.662 | 4.277 | 4.361 | 4.179 | 4.466 | 4.989 | 4.996 | 4.374 | 4.278 | 4.229 | 4.560 |
| EDF [14] | 0.249 | 0.429 | 0.347 | 0.252 | 0.501 | 0.333 | 0.872 | 0.461 | 0.337 | 0.985 | 0.188 | 1.473 | 0.448 | 0.242 | 2.049 |
| D-EDF [15] | 0.312 | 0.545 | 0.425 | 0.337 | 0.682 | 0.328 | 0.966 | 0.417 | 0.345 | 1.024 | 0.304 | 2.047 | 0.567 | 0.488 | 2.249 |
| RiEMann (Ours) | 0.053 | 0.066 | 0.069 | 0.056 | 0.068 | 0.101 | 0.120 | 0.117 | 0.099 | 0.122 | 0.079 | 0.159 | 0.098 | 0.082 | 0.197 |

trained policy on T and four extra settings: 1) New Instance (**NI**): the target object will be replaced by different objects of the same category; 2) New Poses (**NP**): the initial and target poses of target objects will be under SE(3) transformations on the table; 3) Distracting Objects (**DO**): there will be additional distracting objects in the scene; 4) **ALL**: the combination of NI, NP, and DO.

We design three tasks for evaluation. More detailed task descriptions can be found in A.5: 1) *Mug on Rack*: For **T**, we put a mug on the left-down quarter of the table, and put the rack on the right-down quarter of the table, allowing them to rotate within 90 degrees only around the z-axis randomly. For **NI**, we use a new mug and keep the rack the same. For **NP**, we allow the mug and the rack on any quarter of the table, and rotate the mug along all 3 axes with arbitrary degrees; 2) *Plane on Shelf*: This task is designed for the evaluation of objects with complex geometry, as evaluated in [43]. The setting of **T**, **NI**, and **NP** are the same as above; 3) *Turn Faucet*: This is an articulated object manipulation task and the robot must predict SE(3)-equivariant opening direction to turn the faucet. For **NP**, we rotate the faucet along the $z$-axis, and also change the initial position of the handle.

## 5.2 Baselines and Metrics

We compare RiEMann with four baselines: 1) PerAct [37]: a point-cloud-based imitation learning method. This is for the comparison between the SE(3)-equivariant method and the non-SE(3)-equivaraint method. We perform SE(3) data augmentation for PerAct; 2) R-NDFs [13]: the strongest SE(3)-equivariant method that rely on NDFs [12] and Vector Neurons [16]; 3) EDFs [14]: an SE(3)-equivariant baseline that use SE(3)-transformers for manipulation tasks. This is for the comparison between directly regressing target poses (ours) and field matching on type-$l$ fields. Note EDFs require segmented point clouds of the scene and the robot end-effector respectively. Here we manually extract the end-effector point cloud out from the scene for EDFs; 4) D-EDFs [15]: the SOTA SE(3)-equivariant baseline for object rearrangement tasks. This is for the comparison of performances as well as training and inference speeds.

We report the success rates of all methods in all five settings (**T**, **NI**, **NP**, **DO**, **ALL**) with the average success rates across 20 different random seeds. The success rates are the total success rates of the two stages of each task. Additionally, to better evaluate the quantitative results and eliminate the influences of policy-irrelative factors, we propose to evaluate the SE(3) geodesic distance [44] of the predicted target pose $\hat{\mathbf{T}}$ and the ground truth $\mathbf{T}$: $\mathcal{D}_{geo}(\mathbf{T}, \hat{\mathbf{T}}) = \sqrt{\left\|\log\left(\mathbf{R}^\top\hat{\mathbf{R}}\right)^\vee\right\|^2 + \|\hat{\mathbf{t}} - \mathbf{t}\|^2}$, where $\vee$ is the logmap. We report $\mathcal{D}_{geo}$ in the same manner as success rates.

## 5.3 Results

Table 1 and Table 2 show the success rates and $\mathcal{D}_{geo}$ respectively. We can see that PerAct [37] generally performs worse than SE(3)-equivariant methods, which shows using equivariant networks is better than doing SE(3) data augmentation for a non-equivariant backbone. NDFs [13] with Vector Neurons [16] totally fail on unsegmented point cloud input, which shows the necessity of using local SE(3)-equivariance modules. EDFs [14] and D-EDFs [15] achieved mediocre results in both

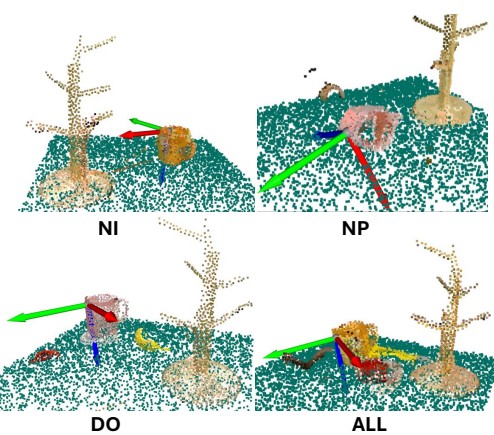
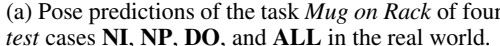

(a) Pose predictions of the task *Mug on Rack* of four *test* cases **NI**, **NP**, **DO**, and **ALL** in the real world.

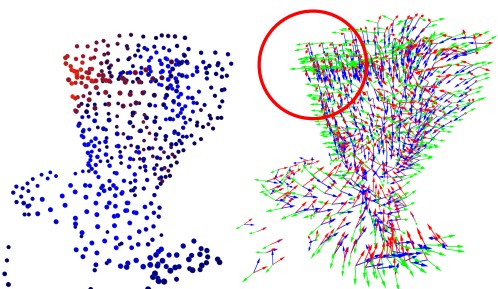

(b) The $\mathbf{B}_{ROI}$ and the local SE(3)-equivariant feature visualization of the **ALL** test cases. The left shows the position (one type-0 vector) field predicted from $\phi_1$. The right shows the orientation (three type-1 vectors) fields from $\phi_2$. We can see that type-1 vectors away from the target position are different from the correct orientation, which shows the necessity of using $r_2$ (the red circle).

Figure 5: Test pose predictions and feature visualization of the real-world task *mug-on-rack*.

success rate and $\mathcal{D}_{geo}$, which shows that they can generalize to local SE(3) transformations of the target objects to a certain degree. However, their performances are worse than RiEMann, especially in $\mathcal{D}_{geo}$, which shows that the field-matching mechanism performs worse than end-to-end policy training. Meanwhile, they cannot support any articulated object manipulation tasks because their output space contains only one target pose. Instead, RiEMann consistently achieves competitive results in success rates significantly better results in $\mathcal{D}_{geo}$ (reducing the average $\mathcal{D}_{geo}$ by 68.6%) under all test settings, which shows the superiority of our end-to-end learning paradigm. Note a $\mathcal{D}_{geo} = 0.05$ means an error around $1cm + 2°$, which is low enough for a successful manipulation. We also visualize the four test case pose predictions in Figure 5a and the local SE(3)-equivariance features in Figure 5b. We can see that the position network $\psi_1$ can attend to the correct region (the rim of the mug), and the orientation network $\psi_2$ can predict correct and consistent rotation matrices around points on the rim of the mug, which shows the effectiveness of RiEMann under local SE(3)-transformations of target objects and new instances, and resist the distraction of other objects.

## 5.4 Real World Evaluation

Finally, we evaluate RiEMann in two real-world experiments: *Mug on Rack* and *Plane on Shelf*, as illustrated in Figure 4. We use four RealSense D435i cameras for point cloud fusion and a Franka Panda arm for execution. We provide 10 demonstrations for each task. We show quantitative results of success rates of pick (P) and full task (A) respectively of each task in Table 5. We can see that the real-world performance of RiEMann is generally the same within the simulation, which shows RiEMann can resist the relatively low-quality point cloud input from the real world. The performance of **NP** is notably suboptimal, primarily due to the partial absence of the object's point cloud in real-world scenarios. This deficiency stems from the cameras' inability to capture the lower section of the object, resulting in a missing portion that faces downwards. Consequently, in both **T** and **NP**, these specific parts exhibit noticeable disparities, as illustrated in Figure 7 in Appendix A.7.1. Please also check supplementary videos for more results and the real-time following video.

## 6 Discussion and Future Works

This paper proposes RiEMann, an efficient and near real-time SE(3)-equivariant robot manipulation framework without point cloud segmentation. One limitation is that equivariant models usually bring more computation and memory costs, which hinders their application in large-scale robot learning scenarios. Another limitation is that RiEMann does not perform well for occluded point cloud input or symmetric objects. Future works can focus on these two problems, and seek to apply RiEMann on other tasks that require SE(3)-equivariant outputs such as forces.

**Acknowledgments**

We would like to thank Tianren Zhang, Yuanchen Ju, and Chenrui Tie for their helpful discussions.

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

# A  Appendix

## A.1  RiEMann Trainig Algorighm

Following the notations in Section 4.2, the training algorithm of RiEMann is summarized as follows:

---
**Algorithm 1** RiEMann Training
---
**Input:** Demonstrations $\{(\mathbf{P}_i, \mathbf{T}_i)\}_{i=1}^M$, initialized models $\phi$, $\psi_1$, $\psi_2$, hyperparameters $r_1$ and $r_2$, epochs $n$.
1: **for** $iter = 0$ to $n - 1$ **do**
2:     Sample a batch of $m$ demonstrations $\{(\mathbf{P}_i, \mathbf{T}_i)\}_{i=1}^m$, where $\mathbf{T}_i = (\mathbf{R}_i, \mathbf{t}_i)$
3:     Predict the saliency map $\mathbf{f}_s(x) = \phi(x), x \in \mathbf{P}_i$
4:     Get $x_{t1}$ by doing weighted sum on $\mathbf{P}$ with the softmax weight from $\mathbf{f}_s(x)$
5:     Get $\mathbf{B}_{ROI}$ centered on $x_{t1}$ with radius $r_1$
6:     Predict $\mathbf{f}_t(x) = \psi_1(x), \mathbf{f}_R(x) = \psi_2(x), \forall x \in \mathbf{B}_{ROI}$
7:     Get $\hat{\mathbf{t}}$ as the weighted position of $\mathbf{f}_t(x)$ and get $\widehat{\mathbf{R}}$ by mean pooing on $\mathbf{f}_R(x)$ on points centered at $\hat{\mathbf{t}}$ with the radius $r_2$
8:     Normalize each type-1 vector of $\widehat{\mathbf{R}}$
9:     Update $\phi$, $\psi_1$, and $\psi_2$ with $\mathcal{L} = \sum_{i=0}^m [\sum_{j=1}^N (\mathbf{t}_i - \hat{\mathbf{t}}_i)^2 + \sum_{k=1}^{N_B} ((\mathbf{t}_i - \hat{\mathbf{t}}_i)^2 + (\mathbf{R}_i - \widehat{\mathbf{R}}_i)^2)]$
10: **end for**
**Output:** Trained models $\phi$, $\psi_1$, and $\psi_2$

---

## A.2  Iterative Modified Gram-Schmidt Orthogonalization

We use Iterative Modified Gram-Schmidt Orthogonalization [41] to make the outputted rotation matrix $\hat{\mathbf{R}}$ legal. IMGS works much more stable than the vanilla Gram-Schmidt Orthogonalization. The algorithm is summarized as follows.

---
**Algorithm 2** Iterative Modified Gram-Schimidt
---
**Input:** $\hat{\mathbf{R}}$ that contains column vectors $v_0$, $v_1$, and $v_2 \in \mathbb{R}^3$
1: **for** $iter = 1$ to $2$ **do**
2:     **for** $i = 0$ to $2$ **do**
3:         $u_i = v_i$
4:         **for** $j = 0$ to $i - 1$ **do**
5:             $v_i = v_i - \frac{\langle v_i, u_j \rangle}{\langle u_j, u_j \rangle} u_j$
6:         **end for**
7:         $u_i = v_i$
8:     **end for**
9: **end for**
**Output:** A legal rotation matrix $\hat{\mathbf{R}}$ that contains updated column vectors $v_0$, $v_1$, and $v_2 \in \mathbb{R}^3$

---

## A.3  Proofs of Theories

First, let's review the definition SE(3)-equivariance on our point cloud $\mathbf{P}$. Given an outputted vector field $\mathbf{f}_{out}(x) = \bigoplus_{i=1}^n \mathbf{f}^i(x), \forall x \in \mathbf{P}$ from SE(3)-transformer [17] where $n$ is the total types of vectors, they are SE(3)-equivariant that means:

$$\mathbf{D}_l(\mathbf{R})\mathbf{f}_l(x) = \mathbf{f}_l(Tx), \forall x \in \mathbf{P}, T = (\mathbf{R}, \mathbf{t}) \in SE(3), l \in n. \tag{5}$$

where $\mathbf{D}_l(R)$ is the Wigner-D matrix. For entities living in the usual 3D physical world, the angular momentum quantum number $j$ of the Wigner-D matrix is 1, thus for vectors with $l = 1$, we have:

$$\mathbf{D}(\mathbf{R}) = e^{-im'\alpha} d_{m',m}^j(\beta) e^{-im\gamma}, \tag{6}$$

where $\alpha, \beta, \gamma$ are the Euler angle representation of $\mathbf{R}$ that satisfies $\mathbf{R} = \mathbf{R}_z(\alpha)\mathbf{R}_x(\beta)\mathbf{R}_z(\gamma)$, $m, m' \in \{-1, 0, 1\}$, and $d^j_{m',m}(\beta)$ is the matrix element. Since $\mathbf{D}(\mathbf{R})$ is also a unitary matrix, we can find a set of basis $[v_0, v_1, v_2]$ that makes $\mathbf{D}(\mathbf{R}) = \mathbf{R}$.

For type-0 vector fields, $\mathbf{D}_0(\mathbf{R})$ is a one-dimensional identical scale factor 1. Let's begin our proofs.

**Theorem 1** *Rotation matrices, represented by three type-1 vectors, are SE(3)-equivariant parameterization of rotation actions.*

*Proof:* For a rotation matrix $\mathbf{R} = [v_0, v_1, v_2] \in \mathbb{R}^9$, where $v_0$, $v_1$, and $v_2$ are the three column vectors, we use three type-1 vectors to represent these three column vectors. Thus the output vector of the network is as follows:

$$\mathbf{f}_{out}(x) = \bigoplus_{i=1}^{3} \mathbf{f}_i^1(x), \forall x \in \mathbf{P}. \tag{7}$$

When the input point cloud is transformed by a SE(3) transformation $\mathbf{T} = (\mathbf{R}, \mathbf{t})$, the rotation matrix representation of the target object also transforms by $\mathbf{R}$, that is $[v_0', v_1', v_2'] = \mathbf{R}[v_0, v_1, v_2]$. According to Equation 5 and 6, the outputted type-1 vector fields are transformed by $\mathbf{D}_1(\mathbf{R}) = \mathbf{R}$, thus we have:

$$[v_0', v_1', v_2'] = \mathbf{D}_1(\mathbf{R})[v_0, v_1, v_2] \tag{8}$$

∎

**Theorem 2** *There is no SE(3)-equivariant vector field representation for Euler angle, quaternion, and axis-angle.*

*Proof:* We use proof by contradiction to prove theorem 2. Consider the example illustrated in Figure 2.

1) For quaternion, we define a quaternion $q = [\cos(\theta/2), \sin(\theta/2)u_i, \sin(\theta/2)u_j, \sin(\theta/2)u_k]$ where $\theta$ is the rotation angle and $[u_i, u_j, u_k]$ is the rotation axis. For the initial pose, we have $q = [1, 0, 0, 0]$. For the end pose, we have $q' = [\frac{\sqrt{2}}{2}, \frac{\sqrt{2}}{2}, 0, 0]$, thus:

$$q' = q + [\frac{\sqrt{2}}{2} - 1, \frac{\sqrt{2}}{2}, 0, 0]. \tag{9}$$

There are two options for quaternion type-$l$ parameterization: 1) using four type-0 vectors; 2) using one type-1 vector and one type-0 vector. For both cases, the $+[\frac{\sqrt{2}}{2} - 1, \frac{\sqrt{2}}{2}, 0, 0]$ operation cannot be represented by a Wigner-D matrix. Thus there is no SE(3)-equivariant vector field representation for quaternion. Axis-angle can be proven in the same way.

2) For Euler angles, we define an Euler angle as $E = (\alpha, \beta, \gamma)$. For the initial pose, we have Euler angles equal to $E = (0, 0, 0)$. For the end pose, we have Euler angles equal to $E' = (-\frac{\pi}{2}, 0, 0)$. Thus we have:

$$E' = E + [-\frac{\pi}{2}, 0, 0]. \tag{10}$$

There are two options for Euler angles' type-$l$ parameterization: 1) using three type-0 vectors; 2) using one type-1 vector. For both cases, the $+[-\frac{\pi}{2}, 0, 0]$ operation cannot be represented by a Wigner-D matrix. Thus there is no SE(3)-equivariant vector field representation for Euler angles.

∎

## A.4 Training Details

### A.4.1 Point Cloud Preprocessing

In the real-world experiments, we perform point cloud voxel downsampling before feeding the point cloud into the network with the voxel size equal to $1cm$ for the *Mug on Rack* task and $2cm$ for the task *Plane on Shelf*.

After this, we perform color jittering by adding Gaussian noise on each point's color with a standard variance equal to 0.005. We also perform random color dropping that replaces 30% points' color to zero, and HSV transformation that randomly transfers the hue, saturation, and brightness of each point by 0.4, 1.5, and 2 times respectively.

### A.4.2 Implementation Details

For RiEMann, We use $r_1 = 0.2m, r_2 = 0.02m$ for all the tasks in the simulation. We use $r_1 = 0.16m, r_2 = 0.02m$ for the real world *Mug on Rack* task, and $r_1 = 0.2m, r_2 = 0.02m$ for the *Plane on Shelf* task. Other network hyperparameters of RiEMann are listed in Table 3. We do not use any kind of prior knowledge for the training of RiEMann such as object segmentation, pertaining, or pose augmentations. We also set the robot to first reach some pre-defined pre-grasp pose (e.g., above the mug) and a pre-place pose (e.g., in front of the rack) to eliminate the unnecessary influence of the motion planners.

The full training of RiEMann takes 200 epochs on a single NVIDIA A40 with a batch size of 4 and a learning rate of 1e-4 for each network module $\phi$, $\psi_1$, and $\psi_2$. However, we find that for *Mug on Rack*, they only need about 50 epochs to converge, which takes about 47 minutes.

Table 3: Network hyperparameters of RiEMann.

|  | Network Layer | Max Type-$l$ | Head Number | Channels | Message Passing Distance |
|---|---|---|---|---|---|
| $\phi$ | 4 | 4 | 1 | 8 | 0.1m |
| $\psi_1$ | 4 | 3 | 1 | 8 | 0.07m |
| $\psi_2$ | 4 | 4 | 1 | 8 | 0.07m |

For PerAct [37], the language descriptions of our tasks are: *Put the mug on the rack*, *Place the plane on the shelf*, *Turn the faucet*. We follow the original 3D voxel grid size ($100^3$) and the patch size ($5^3$). We use Euler angles as the rotational action representations for PerAct. For fairness, we do not train the gripper action for PerAct. We use 6 self-attention layers for the perceiver Transformer module. The other hyper-parameters are the same with the original paper.

For R-NDF [13], since there is no pre-trained weight for the plane and the faucet, we here pre-train the NDFs using the reconstructed meshes from the point clouds in our demonstrations, and use these model as the NDFs module to run R-NDFs. We observe that R-NDFs fail to accomplish all of the tasks when testing, which shows that R-NDFs cannot perform well without object segmentation, because of the locality requirements of R-NDFs. We also tried to use the original pre-trained weights from the original paper [13] for the task *Mug on Rack*, but we found that the performances were even worse because of the discrepancy of the specific object shapes in the test experiments and the pertaining datasets. Other network hyper-parameters are the same as in the original paper.

For EDF [14]and D-EDF [15], we manually separate the robot end-effector and the grasped object point cloud from the scene rather than setting a series of separate cameras to capture their point cloud. For EDFs, we run the MH for 1000 steps, run the Langevin algorithm for 300 steps, and optimize the samples for 100 steps. We use one query point for picking and three query points for placing. We train EDFs for 200 epochs, the same epochs with RiEMann. For D-EDFs, we train the networks for 1 hour with parallel training of the low-resolution and the high-resolution networks and the energy-based critic network. Other network hyper-parameters are the same as in the original paper.

## A.5 Simulation Experiments

### A.5.1 Detailed Descriptions of Simulation Tasks

For all simulation tasks, the radius of the table is $0.75m$, and we put a Franka Panda robot arm in the center of the table. We divide the table into a semicircle part and two quarter parts and make them different heights with a height difference of $0.1m$. This is designed to conveniently apply SE(3)

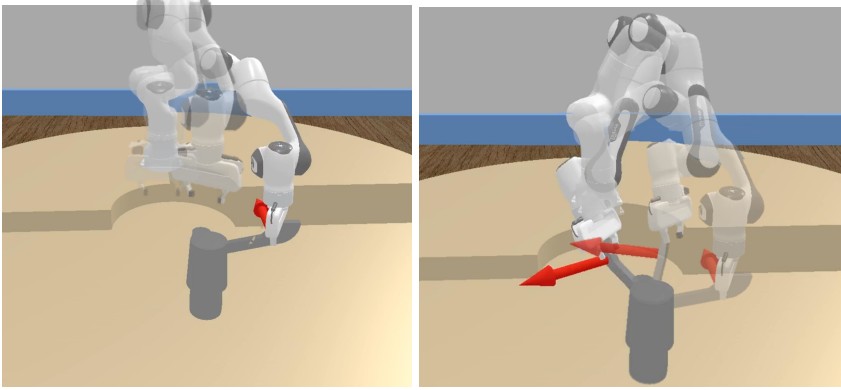

(a) First step: the robot moves to the pre-dicted target pose of the end-effector.

(b) Second step: the robot opens the faucet along the predicted direction (red arrow).

Figure 6: The articulated object manipulation task *Turn Faucet*.

transformations on target objects. We cut the input point cloud into a cube with a side length of 2m centered on the center of the table. Details of different tasks are introduced here:

*Mug on Rack*: A mug and a rack are placed on the table. The robot has to pick up the mug by the rim and then hang it on the rack by the handle. This is the most representative object rearranging task that is also evaluated in [12, 14, 13, 15]. For the training set **T**, we use a mug in blue with a side length of about 17 cm, and a rack with a height of 67cm. The mug must be hung on the highest peg of the rack. For the new instance set **NI**, we use a patterned red mug with a height of about 19cm and a base diameter of about 10cm.

*Plane on Shelf*: A plane model and a box-shape shelf are placed on the table, and the robot has to pick the middle part of the body of the plane and place it on the shelf. For **T**, we use a grey plane model with a length of about 20cm. For **NI**, we use a blue plane with the same size.

*Turn Faucet*: As shown in Figure 6, a faucet is placed on the table, and the robot has to turn on the faucet by first moving to the handle of the faucet and then moving along the opening direction. For **T**, we use the NO. 5004 faucet model in ManiSkill2 [45]. For **NI**, we use the No. 5005 faucet model.

For the *open-faucet* task, we assume that given a target pose $\mathbf{T} = \{\mathbf{R}, \mathbf{t}\} \in SE(3)$ and a target direction $\mathbf{d} \in \mathbb{R}^3$, the robot can accomplish the task by first going to the target pose $\mathbf{T}$ just as done in pick-and-place tasks, and then moving along the target direction $\mathbf{d}$ while keeping the orientation not changed, as illustrated in Figure 6. To encode the extra directional action $\mathbf{d}$, we add another type-1 vector field on the orientation network $\psi_2$, that is: $\mathbf{f}_R = \bigoplus_{i=1}^{4} \mathbf{f}_1^i(x), x \in \mathbf{B}_{ROI}$. The final output directional action $\hat{\mathbf{d}}$ is also calculated through mean pooling on points in the radius $r_2$. Note the policy needs to continuously predict the output direction during the opening process, which shows that RiEMann can capture the local SE(3)-equivariance of the handle part of the faucet.

In this task, we give demonstrations of not only the target pose $\mathbf{T}$, but also the opening direction $\mathbf{d}$ at the first frame. Here there exists two kinds of SE(3)-equivariance: the target pose should be equivari-ant to the pose of the faucet at the first frame, and the opening direction should be equivariant to the handle during the opening process, where the second equivariance requires both local-equivariance and the real-time performance.

### A.5.2 Demonstration Collection

We provide the ground truth pose to a point cloud based motion planner MPlib [46] to generate the demonstration trajectory for training. We transform all point cloud input to the end-effector coor-dinate system. We collect 10 demonstrations for each setting for the evaluation of SE(3) geodesic

Table 4: inference time and the training GPU memory usage (on NVIDIA A40) of RiEMann and D-EDFs on the task *Mug on Rack*.

| | Inference Time | Memory Usage |
|---|---|---|
| RiEMann (Ours) | 0.19s | 11GB |
| D-EDFs [15] | 15.2s | 42GB |

Table 5: Success rates of *Mug on Rack* and *Plane on Shelf* in the real world. Each value is the average of 12 tests.

| | T | | NI | | NP | | DO | | ALL | |
|---|---|---|---|---|---|---|---|---|---|---|
| Task | G | A | G | A | G | A | G | A | G | A |
| Mug on Rack | 1.0 | 1.0 | 1.0 | 1.0 | 0.75 | 0.75 | 0.92 | 0.83 | 0.75 | 0.58 |
| Plane on Shelf | 1.0 | 1.0 | 1.0 | 1.0 | 0.58 | 0.50 | 1.0 | 1.0 | 0.55 | 0.50 |

distance. We manually exclude those situations that cannot support a successful collision-free motion planning trajectory, as well as in the testing cases.

## A.6 Real World Experiments

### A.6.1 Environment Setup

We use a Franka Emika Panda robot arm with four RealSense D435i RGB-D cameras for the real-world experiments, as shown in Figure 4. The cameras are calibrated relative to the robot's base frame. We fuse the point clouds from all four cameras and transform the point cloud into the end-effector frame of the robot for control. We crop the scene to a cube with a side length of 1.5 meters and downsample the point to get 8192 points in the scene.

For real-world tasks, since the point cloud is noisy and usually part-occluded, we do not perform the pose transformation calculation $\widehat{\mathbf{T}} = \mathbf{T}_{place}\mathbf{T}_{object}^{-1}$ for the task, i.e., we directly use the predicted target pose as the final action.

### A.6.2 Detailed Task Descriptions

*Mug on Rack*: The task is similar to the version in the simulation. For **T**, we use a pink mug with a side length of about 10cm. We use a rack with a height of 35cm and a base diameter of about 15cm. We split the table into four equal areas, as in the simulated version of this task. In **T**, we only collect demonstrations in a quarter of the desktop area and let the mug rotate along the z-axis for 90 degrees in a top pose. For **NI**, we use a yellow new mug with a similar size to the pink mug. For **NP**, we let the mug on all table regions and rotate in 3 dimensional with any degree.

*Plan on Shelf*: The task is similar to the version in the simulation. For **T**, we use a blue plane model, and collect demonstrations in the same manner as above. For **NI**, we use a green plane with a similar size. The shelf is a set of discrete racks that can support the plane if it is placed in the correct pose.

### A.6.3 Demonstration Collection

We use the teaching mode of the robot arm to give demonstrations, as illustrated in Figure 1 and in supplementary videos. We transform all the point clouds into the end-effector coordinate system.

## A.7 More Illustrations

### A.7.1 Failure Case Illustrations

We show some failure case point clouds of RiEMann in both tasks in Figure 7. We can see that the lower section of the object is not captured by the cameras, which leads to incomplete geometries.

### A.7.2 Results Illustrations

We here illustrate the results and the features of the *plane-on-shelf* task in Figure 8.

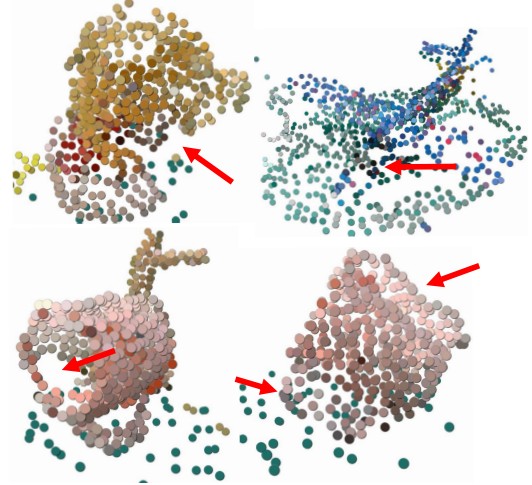

(a) High-quality point clouds in the real world. There are still some flaws in the point clouds, which shows the robustness of RiEMann.

(b) Low-quality point clouds in the real-world. Left up: a big part is missed. Right up: the plane head is missed. Left down: a big part is missed. Right down: noisy points on the mug. These low-quality point clouds can lead to the failure of RiEMann.

Figure 7: Visualization of the low-quality data in the real-world experiments that cause the failure of experiments.

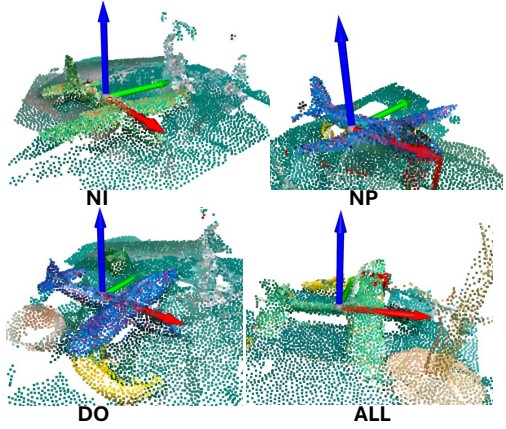

(a) Pose predictions of the task *Plane on Shelf* of four *test* cases **NI**, **NP**, **DO**, and **ALL** in the real world.

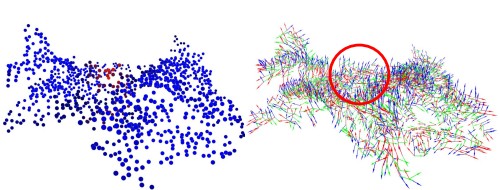

(b) The $\mathbf{B}_{ROI}$ and the local SE(3)-equivariant feature visualization of the **ALL** test cases.

Figure 8: Test pose predictions and feature visualization of real-world evaluations of the *plane on shelf* task.

## A.8    More Results

## A.9    Ablation Studies

### A.9.1    Ablation of Hyperparameters

We test the trained model on the **NP** case of the task *Mug on Rack* in simulation. Results are shown in Figure 9. We can see that with more points in the scene input point cloud, the performance of the model is better, and the same is the number of demonstrations. Our model can achieve competitive results with less than 10 demonstrations. For the hidden layer type-$l$ experiment, we can see that with a higher max type-$l$, the model can work better. However, in practice, higher type-$l$ will extremely

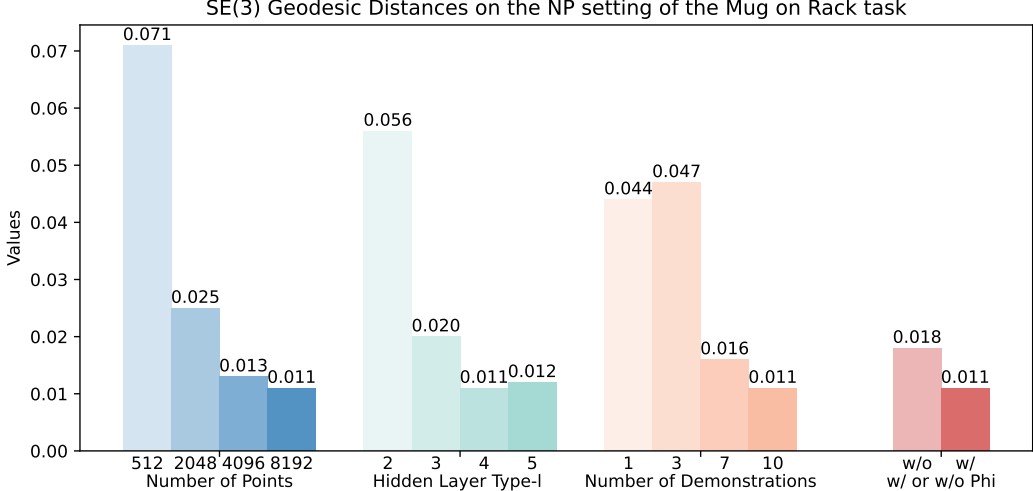

Figure 9: Ablation studies of different hyperparameters of RiEMann. Each value is the average result of 20 random seeds.

increase the computational cost and GPU memory usage. In the *Mug On Rack* task, a network with a maximum number $l$ equals 5 only supports batch size = 1 during training on an NVIDIA A40. Lastly, the last group of Figure 9 shows that the saliency map network can not only reduce the training burden of the policy network but also improve the final pose estimation results.

### A.9.2 Ablation of Radius $r_1$ and $r_2$

Here we add additional experiments to answer this question. We train the *mug-on-rack* task in the simulation with different $r_1$ and $r_2$. Results are in Table 6.

Note in our paper, we choose $r_1 = 0.16m$ and $r_2 = 0.02m$. In the *mug-on-rack* task, the height of the mug is about $0.22m$ and the width of the mug is about $0.20m$. Note $r_2 = 0.01m$ means that there is only one point for $\psi_2$ to perform mean pooling, because we perform point cloud voxelization for the point cloud with a voxel size of $0.01m$.

From the results, we can see that:

- $r_1$ should be large enough ($\geq 0.16m$) to capture all points of the target objects to make sure the position and orientation network can understand the full object geometry. If $r_1$ is less than the radius of the mug, the success rate will drop quickly, and the SE(3) geodesic error will increase quickly. If $r_1$ is too small ($r_1 = 0.05m$), the success rate becomes zero. The reason is that the training becomes very unstable because every time the position and the orientation network may choose different parts of the point cloud for training since the saliency map network is not that strong to give very precise position predictions.

- $r_2$ should capture more than one point ($> 0.01m$) to avoid noisy and accidental errors caused by too few points.

- Although $r_2 = 0.04m$ also works well, it brings more burden for training since the orientation network $\psi_2$ outputs the highest type-$l$ vectors in our whole pipeline, and using fewer points for training is better for reducing the memory usage for $\psi_2$.

- For $r_2 = 0.01m$, the results are significantly worse than other situations in **DO** and **ALL**. This shows that noisy points will influence the results if we only use one point for prediction.

- The worst SE(3)-geodesic error of RiEMann ($r_1 = 0.05m$) is smaller than the R-NDF baseline [13] in Table 2. This is because the saliency map network will restrict the position error to be in a certain range.

Table 6: Success rates and SE(3)-geodesic distances of different $r_1$ and $r_2$ of the *mug-on-rack* task in simulation. Each value is evaluated under 20 random seeds. In our original paper, we choose $r_1 = 0.16m$ and $r_2 = 0.02m$.

| $r_1/m$ | | 0.05 | | | 0.10 | | | 0.16 | | | 0.22 | | | 0.28 | | |
|---|---|---|---|---|---|---|---|---|---|---|---|---|---|---|---|---|
| $r_2/m$ | | 0.01 | 0.02 | 0.04 | 0.01 | 0.02 | 0.04 | 0.01 | 0.02 | 0.04 | 0.01 | 0.02 | 0.04 | 0.01 | 0.02 | 0.04 |
| SR | T | 0.00 | 0.00 | 0.00 | 0.50 | 0.85 | 0.80 | 0.90 | 1.00 | 1.00 | 0.90 | 1.00 | 1.00 | 0.95 | 1.00 | 1.00 |
| | NI | 0.00 | 0.00 | 0.00 | 0.10 | 0.80 | 0.75 | 0.80 | 0.90 | 0.95 | 0.90 | 0.95 | 0.90 | 0.90 | 0.90 | 0.95 |
| | NP | 0.00 | 0.00 | 0.00 | 0.45 | 0.75 | 0.80 | 0.85 | 0.95 | 1.00 | 0.90 | 0.95 | 0.95 | 0.95 | 0.95 | 1.00 |
| | DO | 0.00 | 0.00 | 0.00 | 0.35 | 0.80 | 0.70 | 0.70 | 1.00 | 1.00 | 0.85 | 0.95 | 1.00 | 0.80 | 1.00 | 1.00 |
| | ALL | 0.00 | 0.00 | 0.00 | 0.00 | 0.50 | 0.55 | 0.50 | 0.85 | 0.85 | 0.70 | 0.85 | 0.90 | 0.70 | 0.85 | 0.80 |
| $\mathcal{D}_{geo}$ | T | 1.366 | 1.478 | 1.175 | 0.688 | 0.178 | 0.186 | 0.278 | 0.053 | 0.056 | 0.067 | 0.064 | 0.062 | 0.059 | 0.060 | 0.055 |
| | NI | 1.159 | 1.759 | 1.311 | 0.982 | 0.220 | 0.201 | 0.321 | 0.066 | 0.061 | 0.068 | 0.061 | 0.062 | 0.066 | 0.063 | 0.062 |
| | NP | 1.668 | 1.076 | 1.079 | 0.804 | 0.123 | 0.202 | 0.339 | 0.069 | 0.066 | 0.063 | 0.066 | 0.055 | 0.060 | 0.064 | 0.058 |
| | DO | 1.460 | 1.298 | 1.290 | 0.866 | 0.175 | 0.199 | 0.297 | 0.058 | 0.056 | 0.082 | 0.062 | 0.059 | 0.079 | 0.064 | 0.063 |
| | ALL | 1.982 | 1.333 | 1.077 | 1.390 | 0.797 | 0.639 | 0.427 | 0.071 | 0.070 | 0.079 | 0.077 | 0.068 | 0.088 | 0.079 | 0.072 |

### A.9.3 The Geometry Generalization Ability of RiEMann

The generalization ability of RiEMann on the new geometry comes from the neural network structure and its inductive bias. We think it is the locality property of the message-passing mechanism in the equivariant backbone that brings this geometry-level generalization. We perform the *new instance* experiments because, in related works [14, 15], they found that equivariant networks have a certain degree of geometric generalization ability only with 5 to 10 demonstrations, and we confirmed this point in our experiments. Yes, compared to NDFs [12, 13], our geometry generalization ability is worse because their encoder is trained on a large-scale in-category dataset.

To better demonstrate how strong the generalization ability of RiEMann is on new object geometries, here we add an additional experiment. We gradually deform the mesh of the mug trained in the *mug-on-rack* task, put them into the **NP** setting, and record the corresponding SE(3)-geodesic error of the predictions from the original trained model. We keep the longest side of the mug the same. The results are in Table 7.

Table 7: SE(3)-geodesic distance evaluations of the trained model on the training mug. Each value is evaluated under 20 random SE(3) poses.

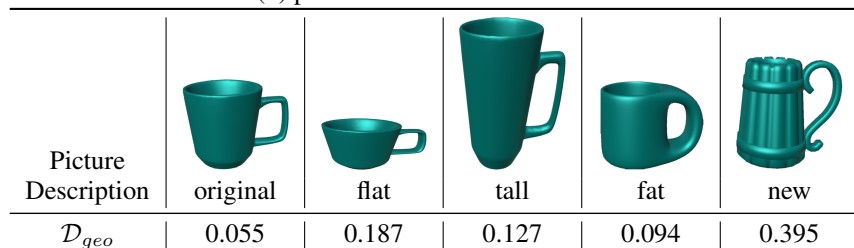

| Picture Description | original | flat | tall | fat | new |
|---|---|---|---|---|---|
| $\mathcal{D}_{geo}$ | 0.055 | 0.187 | 0.127 | 0.094 | 0.395 |

We can see that if the general shape of the mug remains (especially for the rim of the mug), the $\mathcal{D}_{geo}$ remains at a low level, which shows that RiEMann has a certain power of geometry generalization ability. However, if the geometry of the mug is deformed too much, the result becomes worse.

