# OpenReview forum: "RiEMann: Near Real-Time SE(3)-Equivariant Robot Manipulation without Point Cloud Segmentation"
_robot-learning.org/CoRL/2024/Conference — CoRL 2024_

### Official Review · Reviewer_UjoM · 2024-07-10
**The proposed method demonstrates strong generalization ability towards novel scenes, but is limited to uni-model policies.**

**Originality:** 3
**Technical Quality:** 3
**Clarity Of Presentation:** 4
**Potential Impact:** 3
**Recommendation:** 3
**Confidence:** 5

**Review:**

quality

The paper proposes a solid local SE(3) equivariant imitation learning method that brings the benefits of generalizing to out-of-distribution scenes. Moreover, the action regression alleviates expensive SE(3) action inference, compares to previous works that used field-matching or steerable-convolution mechanisms.

clarity

Generally the presentation is clear.  Two small suggestions are: briefly highlight the contributions in the abstract and switch the order of 3.2 and 3.3, so that the group theory backgrounds comes before the equivariance property of the method.

originality

The SE(3) local equivariance and inferring SE(3) action by regression over 3 type0 and 3 type1 features are novel and valid.

significance of this work

-strengths

The proposed method achieves local SE(3) equivariance, thus is robust to novel position, distractors, and novel instance of the class, compares to the training scenes, while maintaining low computation costs.

-weaknesses

The method regresses action pose, thus is incompatible with multi-model actions (i.e., there are 4 orientations to grasp a cube by a parallel jaw gripper).

The method relied on manually tested hyperparameter r1, r2.

The method do not predict pre-pick/ pre-place actions.

Moreover, local equivariance is not always preferred, since more complex tasks requires global information, i.e., to sweep dust using a boom, though the action is related to the gripper and the handle of the boom (local information), the policy needs to reason about the head of the boom and the dust (global information).

**Quality Of The Limitations Section:**

2

**Questions For Rebuttal:**

In line 176-178 states that the rotation matrix is the only SE(3)-equivariant rotation representation. This statement is too strong (maybe wrong). I.e., 5D and 6D representation is also SE(3)-equivariant [1].

[1] Zhou Y, Barnes C, Lu J, Yang J and Li H (2019) On the continuity of rotation representations in neural networks. In: Proceedings of the IEEE/CVF Conference on Computer Vision and Pattern Recognition. pp. 5745–5753.

In Appendix Theorem2, there seems missing the proof for that axis-angle is not SE(3) equivariant.

How can one extend the method to multi-model policies? I.e., how could the method be improved to learn grasping a ball?

**Robotics Focus:**

4

**Summary Of Paper:**

RiEMann is a locally SE(3) equivariant open loop imitation learning framework that predicts gripper pose, given point cloud observation.  RiEMann pipeline includes two stages. First, a saliency network take as input the full point cloud, then selects the region of interest. Second, the position and the orientation are regressed by position and orientation nets, respectively. Since all the networks as well as the action presentations are equivariant, RiEMann achieves end-to-end equivariance.  The local equivariance brings the benefits of generalizing to out-of-distribution scenes. Moreover, the action regression alleviates expensive SE(3) action inference, compares to previous works that used field-matching or steerable-convolution mechanisms.

**Summary Of Recommendation:**

The proposed method achieves SE(3) local equivariance, which is sound and efficient for a subset of tasks. Nevertheless, the method is limited to uni-model policy learning.

---

### Official Review · Reviewer_9VAx · 2024-07-19
**RiEMann demonstrates decent performanace but lacks novellty.**

**Originality:** 2
**Technical Quality:** 2
**Clarity Of Presentation:** 2
**Potential Impact:** 2
**Recommendation:** 2
**Confidence:** 4

**Review:**

**Strengths:**
- The paper is well written.
- The saliency map reduction to reduce the pointcloud size utilized for policy learning is a good design choice to alleviate some of the issues with equivariant models.
- Authors demonstrate the effectiveness of their method on two challenging tasks in the real world.
- Method is applicable to articulated tasks (i.e. turn faucet) which is a under-explored area of robotic manipulation.

**Weaknesses:**
- The authors present two theorems/proofs as contributions but neither are novelties and are commonly discussed topics in group theory.
- The authors use language to describe the group theory behind the equivariance is a bit non-standard. I.e. T-equivariance, type-l vectors, etc.
- Fig.2 is difficult to understand due to the amount of information crammed into a relatively small figure.
- Policy evaluation time should be given in Hz not FPS.
- The authors state that their method is much faster than other equivariant methods but other SO(3) model architectures such as Equiformer-v2 have been shown to be fairly fast.
- Authors state that they ran ablations in the main body but do no report any results. I understand this is due to the page limit but they should not reference them in the main paper unless they report *some* results at the very least in the main body of the work.
- Minor gripe but proposed method acronym (RiEMann ) is both unintuitive and difficult to remember.

**Quality Of The Limitations Section:**

1

**Questions For Rebuttal:**

1. What is T(3)-invariance (Line 153)?
2. Additional details on the model would be appreciated. For example, I assume the saliency network is smaller than the policy network in order to speed up evaluation but these details appear to be missing.
3. How does the speed of this method compare to other more modern SO(3) equivariant model architectures such as Equiformer-v2?
4. Why not compare to Fourier Transporter?

**Robotics Focus:**

4

**Summary Of Paper:**

This paper proposes RiEMann a SE(3) equivaraint open-loop control method for tabletop robotic manipulation. The method boasts an improved evaluation speed by reducing the overall size of the pointcloud used for policy learning. The authors evaluate the method in simulation and in the real world on a small set of tabletop manipulation tasks.

**Summary Of Recommendation:**

While the proposed method does show improved performance when compared to the baselines its overall novelty is a underwhelming. The authors state that their method is the first end-to-end SO(3)-equivaraint model but I do not belive this true and the theoritical contributions have been known for a long time in group theory. Additional model details are needed to properly assses the novely of the model architecture itself.

---

### Official Review · Reviewer_HXLL · 2024-07-20
**Innovative work with some details to be clarified**

**Originality:** 4
**Technical Quality:** 3
**Clarity Of Presentation:** 3
**Potential Impact:** 4
**Recommendation:** 3
**Confidence:** 4

**Review:**

This paper presented a promising object manipulation framework with action field regression. At first glance, the main idea of this paper (action regression) is very simple, and the claim that this simple idea can realize superior performance, generalizability, and efficiency seems too good to be true. I think this paper resolved two major challenges in making this simple idea work.

First, how to map the scene point cloud to an action in a generalizable way? The action prediction is designed as a field regression task, where the translation is estimated through a heatmap regression of the target position, and the rotation is estimated through vector field prediction at the salient area. By casting the action prediction as a dense prediction problem, the framework naturally associates the action to the local part of objects most relevant to the interaction. Importantly, the translation is not regressed as a vector field but as a scalar field,  allowing the generalization benefit from the translation invariance. It requires the translation target to be in the convex hull of the point cloud, which explains why the point cloud of the whole scene instead of the segmented object is input to the network. It also forms a superiority of the proposed method, with less preprocessing required on the input.

Second, how to learn the action field from demonstration efficiently and without explicit labels? There is no ground truth action field to supervise the training. Only a few demonstrations are provided. This paper takes a weighted sum of the dense prediction to supervise the saliency, translation, and rotation estimation. On the one hand, the single action ground truth signal can be propagated to the dense field. On the other hand, the weight naturally modulates the supervision to the most relevant areas of the object.

However, the paper also employed several heuristic designs to make the framework practical. For example, the two scalar field predictions (saliency and translation) seem duplicate and redundant in their function. They both highlight a focus area and constrain the input to this area in downstream processing. Furthermore, there are two predefined radius hyperparameters for the local area selection. These ad-hoc designs show that the action field learning idea needs careful tuning in the implementation details, but I think they are acceptable.

Overall, I think the presentation is clear and easy to follow. The experimental results validate the merit of the proposed framework. However, I have several important questions regarding the clarity of the method and its limitations.

1. How are the demonstration data pair (P_i, T_i) defined? Does the point cloud include the robot arm/gripper, or do you manually remove them for the framework? Since the framework runs near real-time and updates the target continuously, how exactly is the target pose defined? Is it defined with a certain time horizon or with respect to the predefined stages of a certain task? Take the pick-and-place task for example, are there two target ground truth poses for pick and place respectively, or more than that? If it is defined with respect to the stages, do we need to switch stages manually or the network learns it from the point cloud configuration? For the pick-and-place task, the scene remains static before the robot lifts the mug, so how does the robot switch from the "pick" target pose to the "place" target pose?

2. How much data is used for training? Is it only using the data of 5-10 demonstrations in the simulation environment? Are they sliced into the data points per second for training? How many data points are there in total for the network?

3. From my understanding, the success of the pose-equivariance and the action prediction in this framework largely relies on the locality of the learning, given the two-level local area confinement and the SE(3)-transformer's property. Will it restrict the capacity of the network and the potential of this framework to tackle more complicated manipulation tasks?

**Quality Of The Limitations Section:**

3

**Questions For Rebuttal:**

Please see above.

**Robotics Focus:**

4

**Summary Of Paper:**

This paper proposed a robot object manipulation learning method given scene point clouds. The key idea is to directly regress an action field on the point cloud, as opposed to pose estimation from feature matching optimization, so that the inference efficiency can be largely improved. This paper claims to be the first near real-time point-cloud-based object manipulation framework.  SE(3)-equivariant networks are used to enable generalization to pose changes of the target object. Experimental results show good performance and generalizability to pose and instance variations.

**Summary Of Recommendation:**

This work provided innovative insights to the manipulation problem. Method and experiments convincing overall. Some clarification needed.

---

### Author Rebuttal · Authors · 2024-08-09

General Response to All Reviewers:

We thank all reviewers for their valuable and constructive comments, and thanks for the excellent summary of all reviews from Meta Reviewer xPrB. Overall, we appreciate that most reviewers acknowledged the contributions of our paper: **This paper resolved two major challenges in making this simple idea work** by reviewer HXLL, **The saliency map reduction to reduce the pointcloud size utilized for policy learning is a good design choice to alleviate some of the issues with equivariant models** by reviewer 9VAx, and **The SE(3) local equivariance and inferring SE(3) action by regression over 3 type0 and 3 type1 features are novel and valid** by reviewer UjoM.

We address the concerns of each reviewer through corresponding comments together with the rebuttal file. This file includes:
1. equiformerv2.zip: code for using equiformer-v2 as comparison. (for Q6 of reviewer 9Vax)
2. symmetrical_example.png: picture of a symmetrical object. (for Q4 of reviewer UjoM)

---

### Decision · Program_Chairs · 2024-09-04

**Decision:**

Accept

**Comment:**

PRE REBUTTAL:

High level summary of reviews:

Strengths:

- SE(3)-equivariant networks enable generalization to pose changes, distractors and novel instances of the class. The method was called out as being relatively novel.
- The method is well-written and easy to follow (with some caveats about clarity of exposition, see below).
- RiEMann highlighted as maintaining low computation cost (via reduction in point cloud size).
- The experimental results validate the merit of the proposed framework and demonstrate the effectiveness of the method on two challenging tasks in the real world.

Weaknesses:

- Some technical details and heuristic designs in the framework need further clarification and justification. See specific reviewer feedback for details.
- The method relies on manually tested hyperparameters and does not predict pre-pick/pre-place actions, limiting its scope.
- Local equivariance may not be suitable for more complex tasks that require global information, such as sweeping dust with a broom (see reviewer UjoM’s comment).
- Reviewer 9VAx questions whether certain theoretical contributions, like the theorems on SE(3)-equivariance, are not novel and are actually well-established in group theory discussions. Additionally the authors use language to describe the group theory behind the equivariance that is a bit non-standard and this should be fixed.
- The clarity of presentation could be improved, particularly in describing the model details and experimental setup.

POST REBUTTAL:

The authors provided a thorough rebuttal and addressed many of the concerns raised by reviewers. This paper crosses the threshold for acceptance. Please note that Fourier Transporter is unlikely concurrent work (there was at least 3 months between) and the AC agrees this should have been a baseline. However, even with this missing the experimental results are strong enough to justify acceptance. The authors should make sure all the reviewer suggested changes (and their author's own clarifications) are reflected in the final manuscript for the camera ready.